

# Automatic detection of *Opisthorchis viverrini* egg in stool examination using convolutional-based neural networks

Tongjit Thanchomnang[1], Natthanai Chaibutr[2,3], Wanchai Maleewong[4,5] and Penchom Janwan[2,3,6]

[1] Faculty of Medicine, Mahasarakham University, Maha Sarakham, Thailand
[2] Medical Innovation and Technology Program, School of Allied Health Sciences, Walailak University, Nakhon Si Thammarat, Thailand
[3] Hematology and Transfusion Science Research Center, Walailak University, Nakhon Si Thammarat, Thailand
[4] Department of Parasitology, Faculty of Medicine, Khon Kaen University, Khon Kaen, Thailand
[5] Mekong Health Science Research Institute, Khon Kaen University, Khon Kaen, Thailand
[6] Department of Medical Technology, School of Allied Health Sciences, Walailak University, Nakhon Si Thammarat, Thailand

Corresponding author
Penchom Janwan,
pair.wu@gmail.com

## ABSTRACT

**Background:** Human opisthorchiasis is a dangerous infectious chronic disease distributed in many Asian areas in the water-basins of large rivers, Siberia, and Europe. The gold standard for human opisthorchiasis laboratory diagnosis is the routine examination of *Opisthorchis* spp. eggs under a microscope. Manual detection is laborious, time-consuming, and dependent on the microscopist's abilities and expertise. Automatic screening of *Opisthorchis* spp. eggs with deep learning techniques is a useful diagnostic aid.

**Methods:** Herein, we propose a convolutional neural network (CNN) for classifying and automatically detecting *O. viverrini* eggs from digitized images. The image data acquisition was acquired from infected human feces and was processed using the gold standard formalin ethyl acetate concentration technique, and then captured under the microscope digital camera at 400x. Microscopic images containing artifacts and *O.viverrini* egg were augmented using image rotation, filtering, noising, and sharpening techniques. This augmentation increased the image dataset from 1 time to 36 times in preparation for the training and validation step. Furthermore, the overall dataset was subdivided into a training-validation and test set at an 80:20 ratio, trained with a five-fold cross-validation to test model stability. For model training, we customized a CNN for image classification. An object detection method was proposed using a patch search algorithm to detect eggs and their locations. A performance matrix was used to evaluate model efficiency after training and IoU analysis for object detection.

**Results:** The proposed model, initially trained on non-augmented data of artifacts (class 0) and *O. viverrini* eggs (class 1), showed limited performance with 50.0% accuracy, 25.0% precision, 50.0% recall, and a 33.0% F1-score. After implementing data augmentation, the model significantly improved, reaching 100% accuracy, precision, recall, and F1-score. Stability assessments using 5-fold cross-validation indicated better stability with augmented data, evidenced by an ROC-AUC metric improvement from 0.5 to 1.00. Compared to other models such as ResNet50,

InceptionV3, VGG16, DenseNet121, and Xception, the proposed model, with a smaller file size of 2.7 MB, showed comparable perfect performance. In object detection, the augmented data-trained model achieved an IoU score over 0.5 in 139 out of 148 images, with an average IoU of 0.6947.

**Conclusion:** This study demonstrated the successful application of CNN in classifying and automating the detection of *O. viverrini* eggs in human stool samples. Our CNN model's performance metrics and true positive detection rates were outstanding. This innovative application of deep learning can automate and improve diagnostic precision, speed, and efficiency, particularly in regions where *O. viverrini* infections are prevalent, thereby possibly improving infection sustainable control and treatment program.

## INTRODUCTION

Human opisthorchiasis is primarily caused by *Opisthorchis felineus* and *O. viverrini*, two closely related species of liver flukes that have a global impact on human health. The endemic areas of *O. felineus* are predominantly observed in various regions of Southern and Eastern Europe and Western Siberia, while *O. viverrini* exhibits a distribution across Asian areas in the water-basins of large rivers (*Sithithaworn & Haswell-Elkins, 2003*; *Fedorova et al., 2020*). Numerous opisthorchiasis-related topics have been the subject of extensive research in recent decades. However, *Opisthorchis* spp. infections continue to be prevalent. Humans become infected with liver flukes by consuming an infected tradition of raw, fermented, or undercooked cyprinid fish that contains infectious metacercariae. After ingestion, metacercariae excyst in the duodenum and ascend through the ampulla of Vater into the biliary ducts, where they mature and attach to the mucosa. Symptom development is dependent on worm burden and the duration of infection. The majority of *Opisthorchis* spp. infections are asymptomatic, with fewer patients experiencing abdominal pain in the upper right quadrant, indigestion, diarrhoea, flatulence, and fatigue. Symptoms may be more severe in chronic infections; the infection is associated with several hepatobiliary diseases, including cholangiocarcinoma (*Mairiang & Mairiang, 2003*; *Yossepowitch et al., 2004*), the primary cause of death in untreated patients. The usual approach for diagnosing opisthorchiasis includes integrating the history of residing in or traveling to endemic regions, history of eating parasite-contaminated foods, the patient's symptoms, the physical examination, and laboratory findings. A definitive diagnosis of human opisthorchiasis can be made by detecting the egg from a stool specimen by microscopic examination. Early diagnosis of parasite infection is critical to eliminating opisthorchiasis and preventing serious medical conditions in infected individuals. Typically, suspected cases are required to provide stool samples for microscopic examination. In addition, the diagnosis of opisthorchiasis can be accomplished through the utilization of immuno-based methods (*Gómez-Morales et al.,*

2013; *Worasith et al., 2015*; *Emelianov et al., 2016*; *Taron et al., 2021*; *Sadaow et al., 2022*) and PCR-based methods (*Thaenkham et al., 2007*; *Phadungsil et al., 2021*; *Pumpa et al., 2021*), which have demonstrated accuracy ranging from 70.00% to 94.05% and 41.03% to 98.43%, respectively. Using microscopy, an examiner can manually identify the infecting parasite species by clear morphology and count the parasite's eggs. An anthelmintic drug would be provided as a course of treatment after confirmation of infection (*Sripa, Tangkawattana & Sangnikul, 2017*; *Centers for Disease Control and Prevention, 2018*). However, in overworked clinical-microscopy services, relying on qualified examiners to visually inspect fecal smears is not only time-consuming but also concerned with the possibility of human error due to exhaustion and cognitive stress. In addition, cases of low worm load infection and low intensities of fecal worm eggs output which resulting in frequent misdiagnoses. The combination of testing methods and sample preparation protocols was recommended previously (*Sayasone et al., 2015*; *Charoensuk et al., 2019*). Some opisthorchiasis endemic areas, experience difficulties due to the lack of microscopy professionally trained health staff for parasite egg identification, which delays diagnosis and treatment. As a result, infected individuals may develop progressive diseases and life-threatening conditions, and an increase in the number of community members who can be carriers results in high-risk regions.

In the past ten years, machine learning (ML) in medical applications has increased for all tasks aimed at advancing healthcare (*Topol, 2019*; *Kumar & Mahajan, 2020*), drug discovery (*Sun et al., 2020*), and disease diagnostics such as diabetic retinopathy, breast cancer, gastric cancer, cardiovascular and carotid arteries, lung disease and Alzheimer's disease (*Cai, Gao & Zhao, 2020*) and neglected tropical diseases (NTDs) (*Vaisman et al., 2020*). These applications have grown in number even more with the gaining popularity of deep learning (DL) techniques. The DL-based object detection framework that has grown prominent in several computer vision assignments is well-suited for image classification tasks and draws interest in diverse fields, including automatically diagnosing harmful human parasites from clinical specimens (*Kumar et al., 2023*). Many studies have been focused on automatically detecting and classifying the egg of medically significant parasites microscopic images such as soil-transmitted helminths (STHs), intestinal helminths, and blood fluke schistosomes by using DL models. *Holmström et al. (2017)* compared the results of digital image analysis using a mobile microscope and commercially available image analysis software WebMicroscope-DL-based algorithms software to the manual labeling of STH eggs in the images by the researchers. The detection sensitivity of *Ascaris lumbricoides* was 100%, that of *Trichuris trichiura* was 83.3%, and that of hookworm eggs was 93.8%. *Yang et al. (2019)* employed a smartphone equipped with a USB Video Class (UVC) microscope attachment and an artificial neural network-based object detection application named Kankanet, which was trained to recognize eggs of *A. lumbricoides*, *T. trichiura*, and hookworm. The results showed that UVC imaging achieved comparable sensitivity and specificity in detecting *A. lumbricoides* to standard microscopy, while Kankanet interpretation was not as sensitive as parasitologist interpretation. *Delas Peñas et al. (2020)* implemented the You Only Look Once version 3 (YOLOv3) model for the automated detection of helminth eggs; the model performed well, correctly identifying

eggs of *Schistosoma* spp., *Ascaris* spp., and *Trichuris* spp. with a sensitivity of 95.31%, 86.36%, and 80.00%, respectively. *Jiménez et al. (2020)* developed the Helminth Egg Automatic Detector (HEAD) system using an artificial intelligence (AI) model and made it accessible for free online. The HEAD platform can detect eleven species of helminth eggs, and it was successfully used by laboratories from nine different countries for the detection and control of helminthiasis. *Butploy, Kanarkard & Intapan (2021)* proposed the convolutional neural network (CNN) classification algorithm for the three types of *A. lumbricoides* eggs: fertilized, unfertilized, and decorticated. The results showed the classification accuracy of parasite eggs is as high as 93.33%. *Dacal et al. (2021)* established the pipeline utilizing smartphone-assisted microscopy and a telemedicine platform to automatically analyze and quantify STH infection using AI models. Their algorithm was trained and evaluated using 51 stool slides containing 949 eggs of *Trichuris* spp. from six patients. The cross-validation evaluation yielded a precision of 98.44% and a recall of 80.94%. *Lin et al. (2021)* trained a MobileNet V2 model using crowdsourced annotations collected using a customized video game to classify the three classes under study (*Ascaris* spp., *Trichuris* spp., and healthy). The results demonstrated that the models trained with these annotations performed comparably to those trained with expert annotations (the Area Under the Receiver Operating Characteristic curve (ROC-AUC) values of 0.928, 0.939, and 0.932 for child, adult, and expert annotations, respectively). *Lee et al. (2022)* developed the Helminth Egg Analysis Platform (HEAP) as a user-friendly microscopic helminth eggs identification and quantification platform. They have pre-trained and built prediction models of 17 parasite egg species. The HEAP platform facilitates users to import their own parasite egg samples and build a prediction model using backend training pipelines. *Lim et al. (2022)* compared the performance of ML segmentation and DL segmentation to detect human intestinal parasite eggs, including *A. lumbricoides*, *Enterobius vermicularis*, hookworm, and *T. trichiura*. Both approaches yielded 97% to 100% accuracy for each parasite species. The CNN based on the ResNet technique outperformed fuzzy c-Means by intersection over a union (IoU) analysis. *Oliveira et al. (2022)* proposed the faster R-CNN model for automating the diagnosis of *S. mansoni* eggs. In this study, transfer learning was utilized, and the proposed solution gained an average precision of 0.76 for an IoU of 0.50. *Ward et al. (2022)* developed an AI-based digital pathology (AI-DP) device to detect STHs: *A. lumbricoides*, *T. trichiura* and hookworms, and *S. mansoni* eggs. A prototype whole slide imaging scanner was embedded in field studies in four countries. The result achieved an average precision of 94.9 ± 0.8% and an average recall of 96.1 ± 2.1% across all four helminth egg species. Only two prior studies briefly mentioned *Opisthorchis* spp. egg detection and classification. *Naing et al. (2022)* used the YOLOv4-Tiny model to identify 34 classes of intestinal parasitic objects in human stools using 50 images of *O. viverrini* eggs for the training and testing process. The model automatically analyzed the *O. viverrini* egg images accurate results with a precision of 97.35%, a sensitivity of 81.48%, and an F1-score of 88.71%. In short, YOLO is a widely recognized real-time object detection system that has gained popularity due to its outstanding speed and accuracy. The initial introduction of the concept can be explained by *Redmon et al. (2016)*, and further variations have been developed, with the most recent

version being YOLOv7. The YOLO algorithm employs regression theory and a CNN as its underlying framework to achieve efficient and accurate detection of various targets. The algorithm in challenging has played a crucial role in many domains such as autonomous driving, video surveillance, face recognition, and remote sensing satellite imaging (*Wang, Bochkovskiy & Liao, 2022*). *Anantrasirichai et al. (2022)* reviewed the ICIP 2022 Challenge on parasitic egg detection and classification in microscopic images. The committee provided 11 classes of parasitic eggs from stool smear samples for this competition using 1,000 images of *O. viverrini* eggs for the training and 250 images for the testing process. From 30 teams that submitted their results, the average top-five detection results for the *O. viverrini* egg images were accurate results with a precision of 89.00%, a sensitivity of 97.00%, and an F1-score of 93.00%. Improving existing diagnostic standards with AI has the potential to close the gap in detecting and classifying *Opisthorchis* spp. eggs, and it is applicable anywhere that many factors supporting laboratory testing are still lacking. Our expected AI-based model can be processed on a computer running Ubuntu 16.4 LTS (64-bit) with the following minimum specifications: Intel Core i5-8400; CPU@2.8 GHz; and Memory: 6 GB. The prevalence of this carcinogenic liver fluke infection remains high in the Mekong River region; therefore, comprehensive basic stool examination services and prompt treatment are required for the people of these areas. The main objective of this work is to develop an efficient CNN model for automatically detecting *O. viverrini* eggs in a patient's stool sample from microscopic images.

## MATERIALS AND METHODS

### Stool sample preparation and image capturing

The *Opisthorchis viverrini* egg images datasets are freshly prepared from the two *O. viverrini*-infected patient stool samples from Maha Sarakham province by using a gold standard formalin ethyl acetate concentration technique (FECT) (*Elkins, Haswell-Elkins & Anderson, 1986*). Briefly, two grams of stool from each sample were suspended in 10 ml of normal saline solution. The suspension was centrifuged at $700 \times g$ after being strained through two layers of gauze pad into a 15-ml centrifuge tube. Discarded the supernatant and dissolved the sediment with 7 ml of 10% formalin, mixed well, and allowed to stand for 2 min. Then, 3 ml of ethyl acetate was added, mixed well, and centrifuged at $700 \times g$. Discarded the top three layers and dissolved the sediment with 1 ml of 10% formalin. The *O. viverrini*-positive sample was examined and confirmed under a microscope by the expert parasitologist. Each *O. viverrini* egg at the one clear focused-on target image in each microscopic field of view was photographed from an Olympus DP27 digital camera with an Olympus BX43 microscope (Olympus, Japan) at 400×. We used the program's default for acquisition set (Olympus cellSens Standard software Ver.1.18) by the white balance setting of RGB to (1.383, 1.000, 1.699), respectively, turning-off black balance, turning-off saturation and turning-off contrast enhancement. The resolution of the images was 2,448 × 1,920 pixels and was stored in the Tagged Image File Format.

The study's protocol was approved by the Ethics Committee in Human Research Walailak University, based on the Declaration of Helsinki (WUEC-23-051-01). The study's purpose and procedures were explained to the participants prior to enrolment.

Written informed consent was obtained from all participants before sampling collection. All study participants infected with *O. viverrini* were treated with praziquantel.

## Model training and image recognition

Figure 1A shows our project workflow, which consists of two significant steps: training the model to understand the data and applying an object detection approach to identify objects in images. The methodology for fine-tuning our ML model consists of four integral stages: data acquisition, data augmentation, model selection, and model training and validation.

This study encompasses two distinct tasks: one using a non-augmented dataset and the other using a dataset with augmentation. Each dataset underwent five-fold cross-validation, and was then split in an 80:20 ratio, with 80% for training and 20% for validating the model. After training and validation, another set of 40 images was used for testing.

### Data acquisition

Our training dataset contains 200 images where the ratio of *O. viverrini* egg (class 1) to artifacts (class 0) is 1:1. The artifacts, such as pollen, plant cells, and other non-parasitic entities, were included to enable the model to distinguish between actual parasitic eggs and common confounding in stool examinations. To ensure consistency during the training process, all images were resized to a standard size of $200 \times 200 \times 3$ pixels. The images have been resized using bicubic interpolation to keep their quality and features during scaling. The finalized images were used as the CNN's input. These images were prepared by maintaining the visual information's integrity, which was necessary for successfully training and testing the CNN model. By preserving the necessary features in the images and creating a balanced dataset with various *O. viverrini* egg and artifact images, we ensured the model would have a firm basis for learning and detection. A testing dataset, comprising eggs from different sources yet within similar environmental conditions as the training dataset, was curated for evaluation purposes. This testing image dataset consists of 20 images per class (class 0 and class 1) and was utilized to measure the performance of the model.

### Data augmentation

Due to the limited number of original microscopic images available for training the model, several data augmentation techniques were employed to increase the number and variety of the training set. Data augmentation is a common technique employed in ML, especially for image-based projects, to artificially expand the available dataset and improve the model's robustness and performance. According to Fig. 1B, rotation was the first method of augmentation. Each original image was rotated π/2 (90 degrees), resulting in three additional orientations. This ensured that during training, the model was insensitive to the orientation of the egg or artifact within the image. After rotation, we implemented secondary augmentation techniques, including various filtering, noise addition, and image sharpening techniques. These were applied to each rotated image, increasing the number of training images.
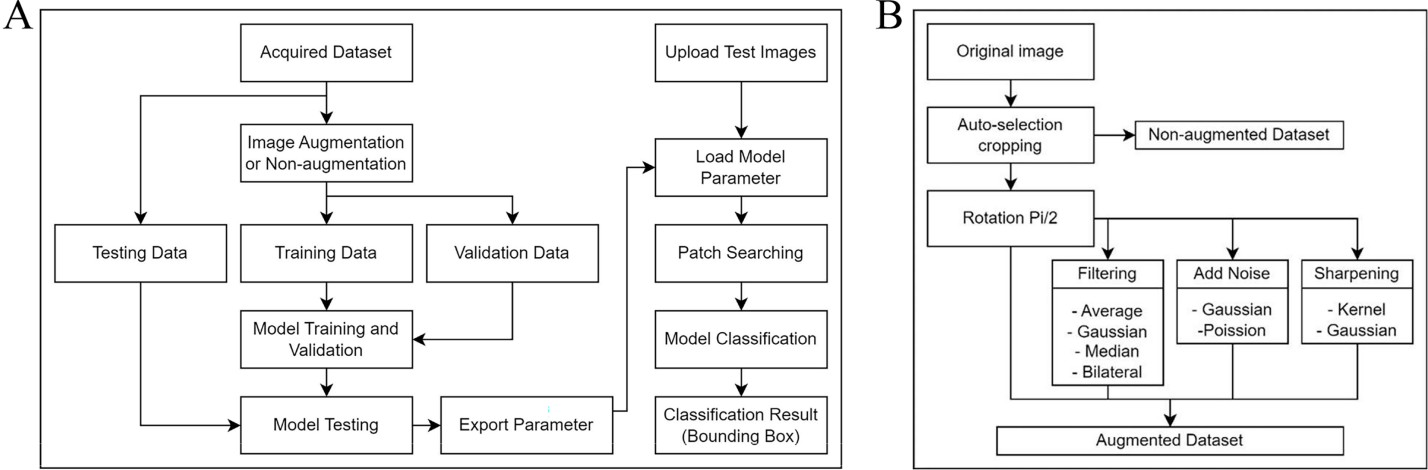

**Figure 1 Process flow diagrams for the experiment.** (A) A comprehensive overview of the workflow involved in our project. The process consists primarily of two integral parts: training of the model, where we condition our system to understand and analyze the data, and implementing the object detection method, which is utilized to identify and locate objects within images. (B) The augmentation algorithm includes a rotation matrix, image filtering, the addition of noise to images, and image sharpening.

**Filtering:** To imitate variations in image quality that may occur in real-world scenarios, we employed four kinds of filters: average, Gaussian, median, and bilateral. All four techniques are major ones used to achieve blurring effects of digital images (*Li et al., 2021*; *Mohanty & Tripathy, 2021*; *Cossio, 2023*) and have a common basic principle: applying convolutional operations to the image with a filter. Each of these filters uniquely modifies the image's appearance, allowing it to diversify the training set and enhance the model's ability to generalize from training data to new images.

**Noise addition:** We included two types of noise, Gaussian and Poisson, to the images in order to further enhance the data and assure robustness against such variations in applications in reality.

**Image sharpening:** Lastly, we applied two image sharpening techniques: the kernel method and the Gaussian kernel method. Image sharpening can help emphasize important image features, enhancing its capacity to detect *O.viverrini* eggs.

The combined augmentation strategies increased our training dataset from 100 to 3,600 images per class (100 original images × 4 rotations × 9 secondary augmentations = 3,600) (Fig. 2). This significantly larger and more diverse dataset allows for more comprehensive training of the CNN, promoting a robust and accurate model capable of identifying *O. viverrini* eggs under various conditions.

## Model selection

Our CNN model is a DL model employed predominantly for image recognition and detection tasks due to its superior ability to recognize image patterns. Figure 3 shows the structure of the model in detail. Our model's design consists of four main parts: input, convolution layer, neural network, and output.

**Input:** The input to the model consists of 200 × 200 pixel color images produced during our sample preparation and data augmentation processes. Before feeding these images into

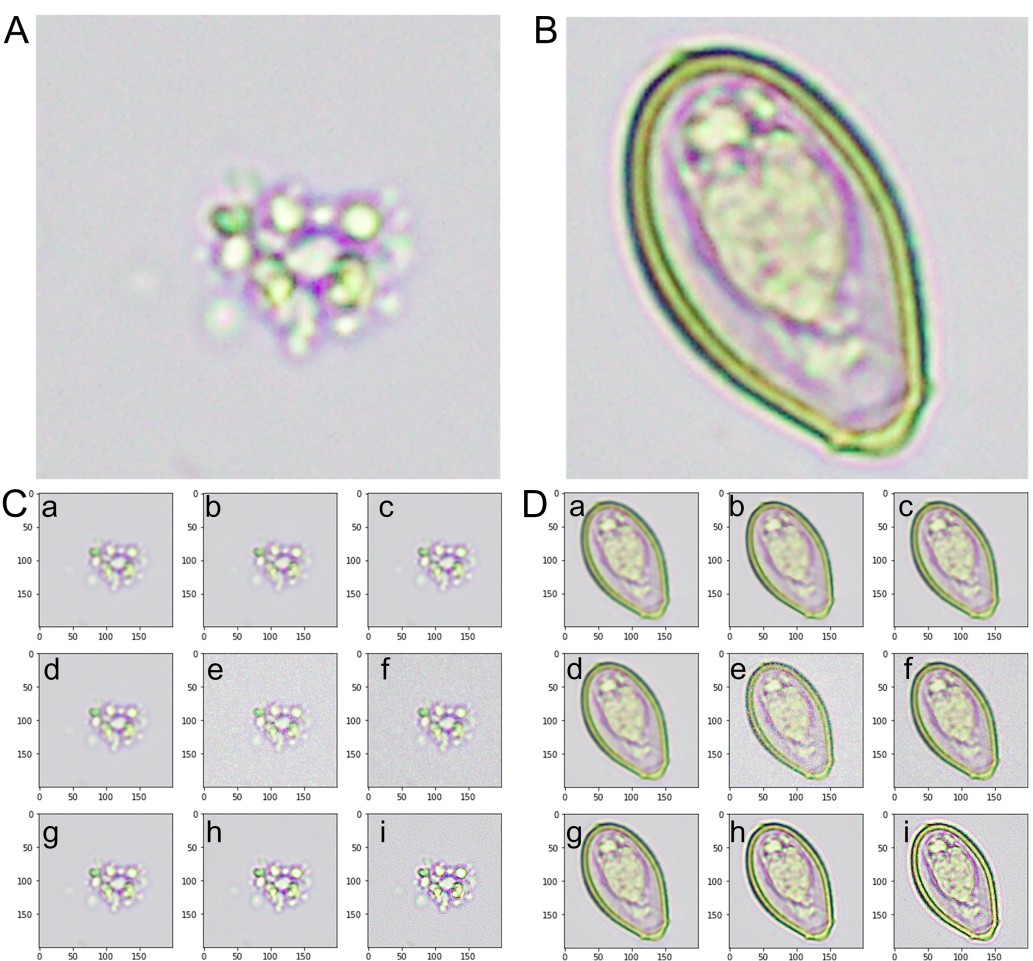

Figure 2 **An instance of images from the training dataset.** (A) The artifact category is identified as class 0. (B) The image of the *Opisthorchis viverrini* egg category is identified as class 1. (C) The representative images from the augmented category were derived from class 0-artifact images. (D) The representative images from the augmented category were derived from class 1-*O. viverrini* egg images. (a) An image with an averaging filter. (b) An image with a bilateral filter. (c) An image with a Gaussian filter. (d) An image with a median filter. (e) An image with Gaussian noise. (f) An image with Poisson noise. (g) An image from a microscopic source with rotation. (h) A sharpened image with a Gaussian kernel. (i) A sharpened image with a kernel.

the convolution operation, they are resized to a uniform size of 128 × 128 pixels using bicubic interpolation to ensure compatibility with the network's input requirements and minimize the required computations. This resizing operation is designed to maintain the images' essential features critical for successful parasitic egg detection.

**Convolution layer:** The convolution layer is the core building block of a CNN. The layer applies a series of different filters to the input images to create a feature map that includes high-level features such as edges, corners, and textures. The feature map is then passed through a non-linear activation function—in this case, a Rectified Linear Unit (ReLU) (*Nair & Hinton, 2010*)—to add non-linearity to the model. Subsequently, a pooling operation is performed to reduce the feature map's spatial size and decrease the model's computational complexity.

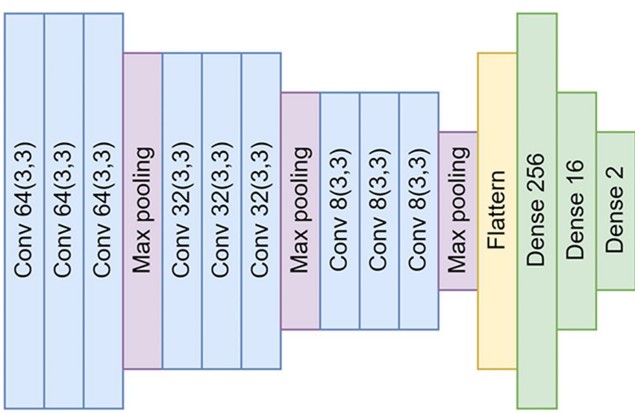

**Figure 3 The architectural design of the proposed convolutional neural networks (CNNs).** The architecture comprises a total of nine convolutional layers and three max-pooling layers. These layers are interconnected by neural networks, which are composed of 256, 16, and ultimately, two nodes for the output layer.

In our proposed neural network architecture, we employ three distinct convolutional layers, each followed by a max-pooling layer. This structure is repeated thrice. Subsequent to the convolutional layers, a flattened layer is utilized to bridge the connection between the convolutional layers and the fully connected neural networks.

**Neural networks:** After passing through the convolutional layer, the feature maps are flattened and fed into a series of fully connected, dense layers. These layers consist of 256, 16, and finally, two nodes for the output layer. They perform high-level reasoning in the neural network, with the final 2-node layer serving to classify the image as either a parasitic egg or an artifact.

**Output:** The output layer uses a softmax activation function to yield a probability distribution over the two classes: *O. viverrini* eggs and artifacts. The model then assigns the input image to the class with the highest probability.

## Model training and validation

For the training and validation of our model, we utilized Google Colab®, which offers the following hardware specifications: a GPU powered by NVIDIA Tesla T4, a CPU featuring Intel Xeon processors operating at 2.3 GHz, 51.0 GB of RAM, and a 320 GB disk.

In our experiments, we utilized a CNN model. The training process was designed to run for up to 200 epochs. However, to ensure the model's efficiency and to prevent overfitting, we incorporated an early stopping mechanism: training would be terminated prematurely if the training loss exceeded a threshold of 0.001. For the optimization process, we selected the stochastic gradient descent (SGD) method. Regarding the model's loss computation, the mean square error (MSE) was chosen as the loss function. Furthermore, to control the speed and stability of our model's learning process, we fixed the learning rate at 0.001. Throughout the training process, the model was taught to distinguish between images of *O. viverrini* eggs and other artifacts. This involved iteratively adjusting the model's weights to minimize the loss function, thereby enhancing its predictive accuracy. The backpropagation algorithm, which is efficient in computing the gradient, was used to

modify the model's weights based on the observed loss and accuracy metrics during training.

The performance of the model was assessed using a suite of evaluation metrics to provide a comprehensive understanding of its predictive capabilities. These metrics included precision, recall, accuracy, and the F1-score, which are defined as follows (*Chicco & Jurman, 2020*):

**Precision:** Precision measures the proportion of true positive predictions out of all positive predictions made by the model.

**Recall:** Recall, also known as sensitivity, measures the proportion of true positive predictions out of all actual positives.

**Accuracy:** Accuracy is the proportion of true predictions out of all predictions made by the model. It provides a general measure of the model's performance.

**F1-score:** The F1-score is the harmonic mean of precision and recall.

In addition to these metrics, we plotted a receiver operating characteristic (ROC) curve and calculated the Area Under the ROC curve (ROC-AUC). The ROC curve illustrates the trade-off between the true positive rate (sn) and false positive rate (1-sp) for every possible classification threshold. The ROC-AUC provides a single metric to summarize the overall performance of the model, with 1.00 representing a perfect model and 0.5 representing a random classifier.

Cross-validation is a statistical method used to estimate the performance of ML models. It is especially useful when we have limited data and want to ensure that our model's performance is not overly dependent on the particular way we split our data into training and testing sets.

In five-fold cross-validation, the dataset is randomly partitioned into five equally (or nearly equally) sized subsets, or "folds". The cross-validation process is then carried out five times, each time using one of the folds as the test set and the other four folds combined as the training set. This way, each data point gets to be in the test set exactly once and in the training set four times.

In this research, we undertook a systematic evaluation of six ML models: our newly proposed CNN and five established architectures, namely ResNet50, InceptionV3, VGG16, DenseNet121, and Xception. The primary aim was to assess and compare their performance capabilities when trained on an augmented dataset, a method known to enhance model generalizability and robustness. To ensure methodological rigor, each model underwent training under uniform conditions. Our evaluation criteria encompassed a suite of performance metrics, including but not limited to accuracy, precision, recall, and the F1-score. Additionally, we scrutinized the efficiency of the training process in relation to the size of the models and the computational resources required. This comprehensive analysis aimed to identify not only the model with the highest accuracy but also the one that optimally balances performance efficacy with resource efficiency.

## Automatic detection of *O. viverrini* egg

Object detection, a fundamental task in computer vision, involves identifying and classifying specific objects of interest within an image. In our context, the objects of interest

are the *O. viverrini* eggs and artifacts. Our object detection algorithm consists of four steps (Fig. 1A): importing testing data and model parameters, applying the patch searching method, interpreting classification result, and visualizing the detected object. The following are details for each step.

In our object detection experiments, we employed a local machine equipped with the following specifications: a GPU powered by NVIDIA GeForce RTX 2060, a CPU with Intel®, Core(TM) i7-10875H operating at 2.30 GHz, 32.0 GB of RAM, and a 256 GB disk storage.

### Import testing data and model parameters

To evaluate the performance of our trained model, the test images are loaded into the object detection algorithm. These images are distinct from those utilized during model training, and we use unprocessed images with approximately 5 million square pixels from the capture system. Additionally, the parameters of the trained model are imported. These parameters are the weights and biases of each model layer, representing the "knowledge" acquired by the model during training. The model uses these parameters to determine the classification of a given image or patch of an image.

### Patch searching method

Due to the high resolution of our images, processing the entire image at once would be computationally expensive and time-consuming. Consequently, we employ a patch-searching technique, also known as a sliding-window approach. This technique divides the image into numerous smaller, uniformly sized patches ($200 \times 200$ pixels) with a sliding-window step size of 50 pixels. The size of these patches is determined based on the expected size of the objects of interest in the images (in our case, the *O. viverrini* eggs and artifacts) and on computational constraints. Each patch is then independently passed through the CNN for classification. The classification process involves the model examining the patch and predicting whether it contains a parasitic egg or an artifact. This method facilitates the efficient scanning of large images and comprehends the spatial distribution of parasitic eggs and artifacts within the image.

### Classification result

The output from the CNN for each patch is a vector representing a probability distribution across two classes: *O. viverrini* egg or artifact. The class with the highest probability becomes the model's prediction for that patch. If the probability exceeds 80%, our algorithm generates a heat mapping image. This process is repeated for every patch in the image, culminating in a comprehensive set of classifications for the entire image.

### Visualizing the detected object

Once all patches have been classified, they are reassembled to recreate the full image. The objects detected by the model (parasitic eggs and artifacts) are then highlighted and labeled within the image for simple visual interpretation. This involves overlaying bounding boxes around the detected objects and accompanying these with labels

indicating the predicted class. This step provides a visually intuitive understanding of the model's performance, showing exactly where it has detected the objects of interest.

This process represents a standard approach to object detection using CNNs. By systematically scanning the image with the patch searching method, the model can accurately locate and classify objects of interest, in this case, the *O. viverrini* eggs and artifacts.

In this study, the output of an object detection algorithm was meticulously compared to bounding box annotations provided by a parasitology expert. This comparison was anchored on the calculation of the IoU metric, a widely recognized standard in object detection accuracy assessment. The IoU metric quantitatively evaluates the overlap between the predicted bounding boxes generated by the object detection model and the ground-truth boxes annotated by the expert. This methodological approach was employed to rigorously assess and validate the performance efficacy of the object detection model, with a particular focus on its precision in identifying and delineating objects of interest within the given dataset.

## RESULTS

In this section, we present and analyze the results derived from the training of our CNN model and its subsequent application in detecting *O. viverrini* eggs from stool samples in our test image dataset. The analysis is divided into two primary phases: the training phase, during which the model learns to distinguish between images of parasitic eggs and artifacts, and the evaluation phase, during which the model's predictive ability on new, unobserved data is evaluated.

### Image datasets

The images used in our study were divided into two classes based on their content: artifacts and *O. viverrini* eggs.

Class 0-Artifact: Fig. 2A presents a sample of our dataset's images categorized as artifacts. Artifacts, in this context, refer to non-*O. viverrini* egg-containing objects or substances found in human stool samples. These include other undigested food particles, plant materials, pollen grains, debris, or other microscopic elements. The variety within this class is considerable, and accurately distinguishing these artifacts from *O. viverrini* eggs poses a significant challenge.

Class 1-*O. viverrini* egg: Fig. 2B presents a collection of the image labeled as *O. viverrini* egg. These eggs' unique morphological features differentiate them from other elements typically found in stool samples. The *O. viverrini* egg is small, oval flask-shaped and yellowish-brown in color with an operculum and distinct shoulders. The size is 22–32 × 12–20 μm in size and contained miracidium inside. Despite their distinctiveness, the task of accurately identifying these eggs among many other components in a stool sample requires a robust and efficient model.

These datasets provide a clear visual representation of the types of images used for training and testing our model. A diverse and representative dataset is crucial for training a model that accurately distinguishes between different classes.

## Augmented dataset

We employed various data augmentation techniques to increase the quantity and diversity of our image dataset and thus enhance the generalization capability of our model. These transformations were applied to the original images from both classes, creating a broader spectrum of features for the model to learn from.

Class 0-Augmented artifacts: Fig. 2C displays a selection of images that were generated from the original artifact images (class 0) using data augmentation techniques. These techniques included rotations, filtering (average, Gaussian, median, and bilateral filter), noise addition (Gaussian and Poisson noise), and image sharpening (kernel and Gaussian kernel). The purpose of augmenting these images was to generate a variety of scenarios and orientations that would aid the model in recognizing artifacts under various conditions and perspectives.

Class 1-Augmented *O. viverrini* eggs: Fig. 2D displays a set of images produced from the original *O. viverrini* egg images (class 1) using the same data augmentation techniques. The augmented images add to the complexity and diversity of the training set, helping the model to better identify *O. viverrini* eggs under various circumstances and in various forms.

These augmented datasets were then used alongside the original images to train our model. By training with a greater variety of images, the model is better able to generalize its acquired knowledge to new, unseen data.

## Model training and validation

The training phase involved the iterative optimization of our model on the augmented training dataset for the course of 200 epochs. We monitored various metrics during this period to evaluate the model's learning progress and predictive performance.

**Loss propagation:** In Fig. 4A, we present the training loss values obtained after 200 epochs for each of the cross-validation folds, specifically from fold 0 through fold 4. It is noteworthy to mention that when the model was trained using an augmented dataset, it consistently exhibited a reduced training loss. Moreover, this loss was observed to be less than 0.001 prior to the completion of the 200 epoch, as can be corroborated by the data displayed in Fig. 4B.

**Accuracy propagation:** A comprehensive examination in Figs. 4C and 4D elucidates the intricate relationship between prediction accuracy and the number of folds employed in cross-validation, particularly emphasizing a 5-fold cross-validation methodology applied to image datasets. We present two distinct scenarios for contemplation: (1) The utilization of non-augmented images unveils a particular trend in accuracy, as evidenced by the overall performance parameters depicted in Fig. 4C, and (2) The application of image augmentation techniques, which yield a consistent accuracy across various folds, as delineated in Fig. 4D.

**Precision-recall metrics:** In a comprehensive evaluation of our ML model, we analyzed its performance using two types of datasets: non-augmented and augmented. When applied to the non-augmented dataset, the model demonstrated an accuracy rate of 50.0%. Its precision, which reflects the proportion of true positive identifications among all

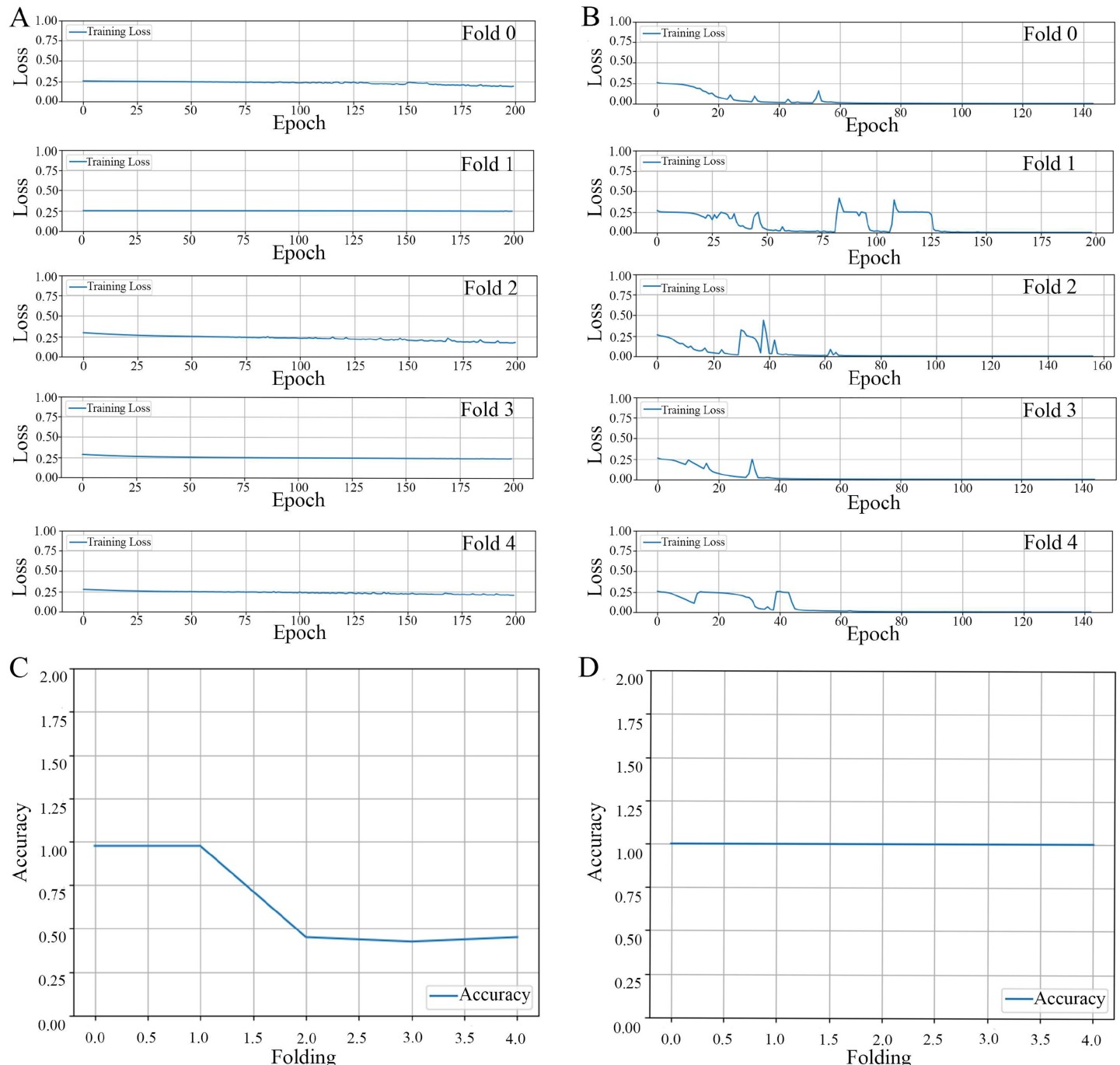

**Figure 4 The results of the model training and validation.** The results of a five-fold cross-validation training loss. (A) Non-augmented image dataset. (B) Augmented image dataset. The relationship between prediction accuracy and the number of folds used in cross-validation for image datasets. (C) Non-augmented image dataset. (D) Augmented image dataset.           

positive predictions, was recorded at 25.0%. The model's recall rate, indicative of the ability to identify true positives from the actual positives, also stood at 50.0%. Furthermore, the F1-score, a harmonized measure of precision and recall, was computed to be 33.0%. Contrastingly, the model's efficacy markedly improved when tested against the augmented

dataset. In this scenario, it achieved exemplary results, registering a 100% score across all key performance indicators: accuracy, precision, recall, and F1-score. This outstanding performance highlights the model's enhanced generalization capabilities when trained on augmented data, enabling it to consistently make correct predictions across all evaluated test cases.

**ROC curve:** Figure 5 illustrates two distinct ROC curves, which graphically represent the model's ability to distinguish between target classes by plotting the true positive rate (sn) against the false positive rate (1-sp) across various classification thresholds. For the non-augmented dataset, the ROC curve demonstrated a robust ROC-AUC value of 0.50 (Fig. 5A), highlighting the model's notable proficiency in classification tasks and implying a high degree of predictive accuracy. Contrarily, the ROC curve for the augmented dataset achieved an optimal ROC-AUC value of 1.00 (Fig. 5B), underlining the model's exceptional discriminative skill and precision in differentiating between positive and negative instances. The ROC-AUC serves as a succinct summary of the model's performance, where a score of 1.00 signifies perfect classification, and 0.5 indicates a capability equivalent to random chance.

This study presents a comparative analysis between the outcomes derived from the conventional model and those obtained using the proposed model. As illustrated in Table 1, the proposed CNN, along with ResNet50, InceptionV3, and DenseNet121, achieved exemplary performance, registering perfect scores in the classification task. The decision to select a model for deployment in object detection tasks was guided by several criteria, notably the model size and its efficacy in processing an augmented dataset. Based on these considerations, the proposed CNN model was identified as the most suitable candidate. This choice was underpinned by its demonstrated capability to efficiently handle complex datasets, thereby ensuring robustness and accuracy in object detection applications.

## Automatic detection of *O. viverrini* egg

After training our model to effectively distinguish between *O. viverrini* eggs and artifacts, we applied it to detect and classify the objects in microscopic images.

Figure 6A illustrates a representative example of a true positive prediction concerning an *O. viverrini* egg. This figure illustrates the successful identification of an *O. viverrini* egg by the model, where the prediction aligns accurately with the actual presence of the egg in the microscopic image. Figure 6B, on the other hand, presents a representative instance of an IoU overlay for an *O. viverrini* egg. In this visual representation, the green bounding box, as annotated by the parasitology expert, delineates the actual location of the egg, while the red bounding box, generated by the model, indicates the predicted location. This overlay method serves not only as a tool for visualizing the congruence between the expert's annotations and the model's predictions but also provides a clear and quantifiable means to assess the precision and accuracy of the model in localizing and identifying *O. viverrini* eggs in microscopic images.

Within the dataset comprising 148 microscopic images, the performance of our model was evaluated based on the IoU metric. According to the data presented in Table 2, 139

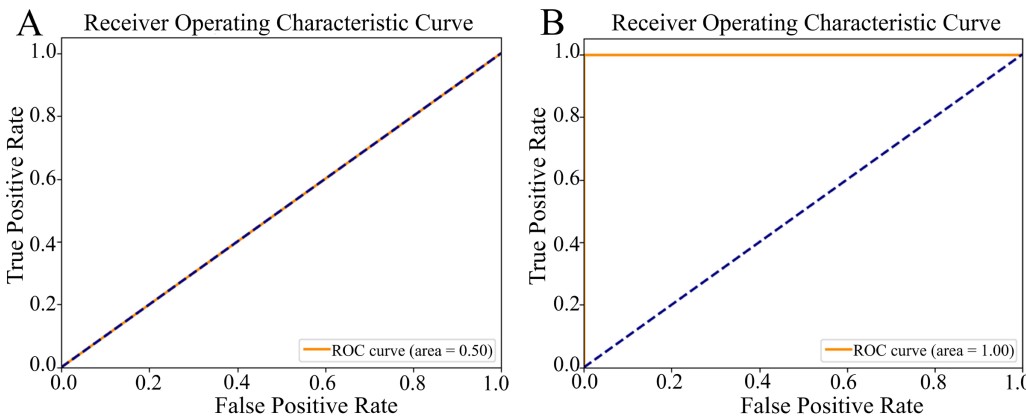

**Figure 5 The Receiver Operating Characteristic (ROC) curves.** (A) The ROC curve of the non-augmented dataset, with the Area Under the ROC curve (ROC-AUC) value of 0.50 indicates random discriminatory ability of the model. (B) The ROC curve of the augmented data, with an AUC value of 1.00 indicates excellent discriminatory ability of the model, demonstrating its effectiveness in distinguishing between the positive and negative classes.               

**Table 1 A comparison of the results from the conventional and proposed models, including attributes data, performance metrics, and file size.**

| Model | Augmented data | Accuracy | Precision | Recall | F1-score | ROC-AUC | File size (KB) |
|---|---|---|---|---|---|---|---|
| Proposed CNN | None | 0.50 | 0.25 | 0.50 | 0.33 | 0.50 | 2,733 |
| Proposed CNN | Yes | 1.00 | 1.00 | 1.00 | 1.00 | 1.00 | 2,733 |
| Resnet50 | Yes | 1.00 | 1.00 | 1.00 | 1.00 | 1.00 | 276,972 |
| InceptionV3 | Yes | 1.00 | 1.00 | 1.00 | 1.00 | 1.00 | 256,730 |
| VGG16 | Yes | 0.50 | 0.25 | 0.50 | 0.33 | 0.50 | 762,594 |
| Densenet121 | Yes | 1.00 | 1.00 | 1.00 | 1.00 | 1.00 | 83,847 |
| Xception | Yes | 0.50 | 0.25 | 0.50 | 0.33 | 0.50 | 244,794 |

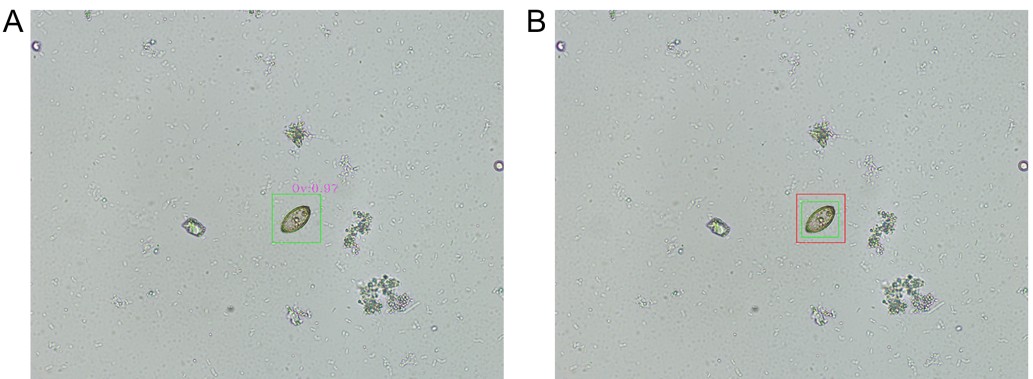

**Figure 6 The representative results of object detection.** (A) A true positive prediction for an *Opisthorchis viverrini* (Ov) egg by the proposed model is illustrated. (B) An intersection over union (IoU) overlay for an *O. viverrini* egg, where the expert's annotation is marked with a green box and the model's prediction is indicated by a red box.               

**Table 2 The results include the intersection over union (IoU) range and image count.**

| IoU | Image count | Average IoU | %Average IoU |
|---|---|---|---|
| <0.5 | 9 | 0.1211 | 12.11 |
| >0.5 | 139 | 0.6947 | 69.47 |

images achieved an IoU score exceeding 0.5, indicative of a high degree of accuracy in detection. Conversely, nine images registered an IoU score below 0.5, encompassing those instances where the model failed to detect any objects. This category includes non-detected images, highlighting areas where the model's detection capability could be improved. The ability of our model to accurately identify and localize eggs, as evidenced by the substantial proportion of correctly detected eggs, underscores its potential utility as an automated tool in the detection and diagnosis of *O. viverrini*. This finding is particularly significant, given the complexity of interpreting microscopic images in parasitology, and suggests that our model could serve as a valuable aid in enhancing diagnostic accuracy and efficiency.

## DISCUSSION

Automated methods in healthcare diagnostics have gained significant attention, with CNNs being one of the most commonly used models for image-based tasks (*Litjens et al., 2017*). In this study, we successfully utilized a CNN to automate the detection and classification of *O. viverrini* eggs in microscopic stool sample images, a task that was typically performed manually by medical technicians using microscopy. Data augmentation is used for improving the performance of a model in medical image analysis by increasing the quantity and variety of data (*Perez & Wang, 2017*; *Garcea et al., 2023*). Our study applied various image augmentation techniques to simulate the various microscopy examination conditions. These techniques consist of rotation, filtering, the addition of noise, and image sharpening. Since microscopic images can be oriented in any orientation, rotations are especially efficient in this context. By rotating the images, we ensure that our model is robust against variations in egg orientation. To simulate various illumination conditions and the presence of debris in human stool samples, filtering and noise addition are used. These techniques also help to mimic the variations and imperfections typically observed in actual microscopic examinations. Additionally, we used sharpening techniques to enhance the edges of the eggs, making it easier for the model to distinguish the parasitic eggs from the background. Similar techniques for augmenting model performance have been implemented in other studies (*Shorten & Khoshgoftaar, 2019*; *Delas Peñas et al., 2020*; *Dacal et al., 2021*; *Lin et al., 2021*; *Naing et al., 2022*), demonstrating their efficacy. By generating a more diverse and representative dataset, we improved the robustness and generalizability of our model, allowing it to manage variations in microscopic imaging conditions that occur in the real world (*Anantrasirichai et al., 2022*). The profound impact of data augmentation on model training and testing is illuminated through a comparative analysis, wherein the data size is

amplified 36 times through augmentation. Engaging in model training without leveraging data augmentation manifests a trajectory characterized by elevated error rates and a sluggish adaptive response to the training process. Conversely, employing data augmentation not only mitigates error rates during training but also fosters a conducive environment for the model to learn effectively. Additionally, the variance in accuracy across different folds of the model is notably distinct when contrasting augmented and non-augmented data. Non-augmented data presents a landscape of uncertainty and variability in accuracy across folds, potentially compromising the model's robustness. Augmented data, on the other hand, perpetuates a consistently high-value accuracy, instilling a stable and reliable aspect of the model's predictive capabilities. Furthermore, when subjected to testing, models trained on non-augmented data achieved an accuracy of 0.50, while those trained with augmented data remarkably reached the pinnacle of accuracy, scoring 1.00. This underlines the pivotal role that data augmentation can play in enhancing model performance, reducing error, and stabilizing accuracy across diverse data folds, thereby fortifying its generalization capabilities in unseen data scenarios. One significant highlight is our approach to separating the background from the objects (parasitic eggs or artifacts) in the images. This class separation is crucial for reducing computational complexity and improving model performance, as it enables the model to focus solely on the relevant features of the eggs and artifacts rather than the extraneous information in the background (*Zheng et al., 2015*). This strategy has been effectively employed in other studies (*Yang et al., 2001*; *Holmström et al., 2017*; *Dacal et al., 2021*; *Lim et al., 2022*) and is consistent with the fundamental practice of the laboratory technician in the field of parasitology, which is to distinguish parasite subjects from misleading artifacts. Secondly, the quality of the images used in our study was significantly enhanced using the gold standard FECT in stool sample preparation. This method has been recognized as one of the most effective stool concentration techniques, resulting in clear images with minimal debris (*Hughes et al., 2017*; *Charoensuk et al., 2019*; *Wattanawong et al., 2021*; *Boonjaraspinyo et al., 2022*; *Perakanya et al., 2022*). More explicit images not only make the detection process more accessible but also contribute to higher accuracy levels in the model's performance.

A recently published version of the World Health Organization's (WHO) 2021–2030 road map for NTDs pointed out that diagnostics are essential for driving NTD-endemic nations toward the challenging goals they have set (*WHO, 2020*). However, most NTD programs rely on the microscopic examination of human specimens, which requires trained personnel and specific expertise. Our findings align with the broader trend of applying DL to detect parasites in humans life-threatening (*Kumar et al., 2023*), which has the potential to close the existing diagnostic standards gap. Malaria, STH infections, and schistosomiasis are of considerable concern to the WHO due to their association with large morbidity and mortality rates, particularly in countries with low-and middle-income countries. Currently, there is a significant research emphasis on the aforementioned three diseases, with the CNN-based DL model being widely adopted for the classification of parasitic microscopic images. Most of these models were developed based on the current state-of-the-art in image classification and object detection. Some notable examples

include the customized CNN models proposed by *Liang et al. (2017)*, *Manescu et al. (2020)*, and *Butploy, Kanarkard & Intapan (2021)*. Additionally, ResNet models have been used, as demonstrated by the works of *Rajaraman et al. (2018)*, *Reddy & Juliet (2019)*, *Nayak, Kumar & Jangid (2020)*, and *Lim et al. (2022)*. YOLO models have also been employed, as demonstrated by the studies conducted by *Delas Peñas et al. (2020)* and *Naing et al. (2022)*. Furthermore, the MobileNet V2 architecture has been utilized, as proposed by *Lin et al. (2021)*. Lastly, the faster R-CNN model has been employed, as demonstrated by *Oliveira et al. (2022)*. The results of the investigation demonstrate that the evaluation of accuracy exceeded 90%. Several studies have offered the idea of applying in-house devices and AI-based software systems specifically designed for diagnosing infections caused by STHs (*Holmström et al., 2017*; *Yang et al., 2019*; *Ward et al., 2022*). Furthermore, other researchers have proposed the advancement of helminth egg analysis as a user-friendly platform (*Jiménez et al., 2020*; *Dacal et al., 2021*; *Lee et al., 2022*). The aforementioned proofs-of-concept demonstrate the potential of their research to effectively detect and classify common helminth eggs in human specimens through the visual examination of digitized images. Despite the remarkable results, it is necessary to remember that successfully implementing the DL model in a real-world clinical setting would involve overcoming several obstacles. Although our model demonstrated high precision and recall, it yielded false negatives. In a clinical context, these could result in missed diagnoses, which would have detrimental effects on the patient's health. Therefore, we suggest the laboratory staff re-examines all opisthorchiasis negative results cases under the microscope for final confirmation. This problem is not specific to our study; other studies, such as *Naing et al. (2022)* and *Anantrasirichai et al. (2022)*, have struggled to reduce false negatives in their models. In future attempts, we planned to collect more samples, including different kinds of parasitic eggs, and employ different sample preparation techniques for use in a real-world clinical setting. Future studies should investigate the application of more advanced ML methods, such as CNN derivatives, ensemble learning, and transfer learning. In the past decade, various CNN architectures have been introduced and modified. These modifications include structural reformulation, reorganization of processing units, network depth, regularization, parameter optimizations, *etc*. The selection of an appropriate model architecture plays a crucial role in enhancing the performance of different applications. The various state-of-the-art CNN architectures such as AlexNet, VGG, GoogLeNet, ResNet, DenseNet, MobileNet, and Inception (*Rajaraman et al., 2018*; *Alzubaidi et al., 2021*). Ensemble learning is a technique in which multiple models are trained on the same dataset. This can often result in superior performance compared to using a single model and reduce the likelihood of selecting an insufficient model. By combining the predictions of multiple models, ensemble methods often achieve higher accuracy and are more robust to varying data and tasks (*Khan, 2022*; *Marques, Ferreras & de la Torre-Diez, 2022*; *Bhuiyan & Islam, 2023*). The transfer learning method, in which a previously trained model is applied to a new but related task, has the potential to enhance the model's performance (*Shin et al., 2016*; *Jameela et al., 2022*; *Ward et al., 2022*). Additionally, it was demonstrated in studies by *Oquab et al. (2014)*, *Yosinski et al. (2014)*, *Rajaraman et al. (2018)*, *Lee et al. (2022)*, *Ward et al. (2022)* that employing

pre-trained models might greatly minimise the needed computational resources. In the future, we will be introducing additional interdisciplinary methods that combine the knowledge and experience of medical professionals with DL-based systems in order to increase the model's efficacy and diversity. In addition, the model will also be deployed on smartphones or portable devices (*Holmström et al., 2017*; *Yang et al., 2019*; *Meulah et al., 2023*), and clinical proof of concept will be conducted for telehealth applications (*Jiménez et al., 2020*; *Dacal et al., 2021*; *Lee et al., 2022*). The concept of distant consulting and remote access to laboratory analysis for field testing is interesting. Regarding this particular context, the suggested approach has the potential to decrease the time, distances, workload of medical technicians and proficiency skills required for the microscopic examination of helminth samples. Consequently, this method could contribute to the accessibility of accurate diagnoses for parasitic infections.

The present study, while offering valuable insights, does contain several limitations. First, the quantity of original images in our dataset is constrained; we possess merely a hundred original images for each class. This limited number could potentially impact the comprehensiveness and robustness of our results. Secondly, a clear distinction arises between FECT images and simple smear images, which are typically used in the routine laboratory in the hospital setting. The clarity evident in FECT images is attributable to the FECT filtering process. This discrepancy may create variances in the interpretation and comparison of results, which should be considered. Lastly, this study's primary focus lies on only one parasitic species, *O. viverrini*, for which our CNN model generates a single-class classification output. However, in the actual FECT samples analyzed, we can discover eggs from other parasite species. Thus, the presence of these additional species may influence the specificity of our findings related to *O. viverrini*, and our results may not be fully applicable in a broader context where various parasitic species coexist. This scenario requires further study by modifying the CNN-based supervised learning model for producing multi-class classification output (*Holmström et al., 2017*; *Delas Peñas et al., 2020*; *Jiménez et al., 2020*; *Anantrasirichai et al., 2022*; *Lee et al., 2022*; *Naing et al., 2022*; *Ward et al., 2022*), which can effectively identify the common parasites that cause disease in humans and provide greater diagnostic usefulness. To the authors' knowledge, the identification of thirty-four classes of intestinal parasites using the YOLOv4-Tiny model by *Naing et al. (2022)* is the greatest number of classes detected to date. However, it is noteworthy that despite utilizing the helminthic worm type solely *O. viverrini* egg data in training our CNN model. A more thorough consideration of the resemblance between the *O. viverrini* egg morphology and that of *Opisthorchis*-like eggs would enhance our research's applicability in examining eggs within the Opisthorchiidae family. This family includes the three major human carcinogenic-related liver flukes: *O. viverrini*, *O. felineus*, and *Clonorchis sinensis*, which are known to be impossible to distinguish between the three egg types under the microscopic examination method (*King & Scholz, 2001*; *Yossepowitch et al., 2004*). Expanding the scope of our research application to encompass these related species within their respective endemic settings would provide additional benefits. Specifically, *O. felineus* is predominantly observed in various regions of Southern and Eastern Europe and Western Siberia, while *C. sinensis* exhibits a wider distribution across

East Asia, with a main presence in China, the Korea, and northern Vietnam (*Sithithaworn & Haswell-Elkins, 2003*; *Fedorova et al., 2020*).

## CONCLUSIONS

No previous studies have emphasized investigating the diagnostic effectiveness of the CNNs model in human opisthorchiasis. This study successfully demonstrated the application of CNNs in automating the detection and classification of *O. viverrini* eggs in human stool samples. Our model showed excellent performance metrics throughout the training phase, with accuracy, precision, recall, F1-score, and ROC-AUC all equal to 1.00, as evidenced by a five-fold cross-validation which suggested stable properties of training from the augmented dataset. Furthermore, during the object detection phase, the model exhibited a high degree of accuracy in its detections, as evidenced by the substantial IoU scores when compared with the labels provided by the expert. This study provides valuable insights into the prospective applications of ML, specifically CNNs, in medical image diagnostics related to parasitology. Our model indicates the viability of automated detection and classification of *O. viverrini* egg, laying the foundation for future research in this field. This provides the way for more efficient and effective diagnostic procedures, which could ultimately lead to improved opisthorchiasis sustainable treatment and control program. Our proposed methodology enhances the efficiency of detecting *O. viverrini* eggs, which traditionally relies on labor-intensive and expertise-demanding microscopic examination. Automation of this assistive process has the potential to alleviate the burden on healthcare professionals and improve the speed and precision of diagnosis. This is especially important in regions where *O. viverrini* infections are prevalent and microscopists with relevant experience may be limited.

## ACKNOWLEDGEMENTS

We acknowledge all participants for their contribution of time and patience in the study. We are grateful to Mr. Pongphan Pongpanitanont who provided data engineering support and substantial comments for improvement.

### Funding

This project is funded by the National Research Council of Thailand (NRCT) and Walailak University: N42A650376. In addition, this research work is financially supported by Walailak University Graduate Research Fund. The funders had no role in study design, data collection and analysis, decision to publish, or preparation of the manuscript.

### Grant Disclosures

The following grant information was disclosed by the authors:
National Research Council of Thailand (NRCT) and Walailak University: N42A650376.
Walailak University Graduate Research Fund.

## Competing Interests

The authors declare that they have no competing interests.

## Author Contributions

- Tongjit Thanchomnang conceived and designed the experiments, performed the experiments, analyzed the data, prepared figures and/or tables, authored or reviewed drafts of the article, and approved the final draft.
- Natthanai Chaibutr analyzed the data, prepared figures and/or tables, authored or reviewed drafts of the article, and approved the final draft.
- Wanchai Maleewong conceived and designed the experiments, performed the experiments, authored or reviewed drafts of the article, and approved the final draft.
- Penchom Janwan conceived and designed the experiments, performed the experiments, analyzed the data, prepared figures and/or tables, authored or reviewed drafts of the article, and approved the final draft.

## Human Ethics

The following information was supplied relating to ethical approvals (*i.e.*, approving body and any reference numbers):

The study protocol was approved by the Ethics Committee in Human Research Walailak University (WUEC-23-051-01, approved 20 February 2023) in accordance with the Declaration of Helsinki.

## Data Availability

The codes are available at figshare: Thanchomnang, Tongjit; Chaibutr, Natthanai; Maleewong, Wanchai; Janwan, Penchom (2023). Codes. figshare. Software. https://doi.org/10.6084/m9.figshare.24712350.v1.

The data is available at figshare: Thanchomnang, Tongjit; Chaibutr, Natthanai; Maleewong, Wanchai; Janwan, Penchom (2023). Data. figshare. Figure. https://doi.org/10.6084/m9.figshare.24712389.v1.

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
