# Peer review of "Automatic detection of Opisthorchis viverrini egg in stool examination using convolutional-based neural networks"

_PeerJ, doi:10.7717/peerj.16773_

## Round 0.1 · original submission · Major Revisions

The review process is complete, and three reviews are included at the bottom of this letter (also see the annotated file). All reviewers and I agree that your manuscript deserves to be published since the authors introduce a machine-learning model aimed at identifying Opisthorchis viverini eggs in digital images. While the experimental design and findings are original and relevant to the field, we identified some concerns and flaws that must be considered in your resubmission. Please, provide rigorous details related to the model to attend Reviewer #1 concerns.

Reviewer 1 ·

Basic reporting

The authors of the study present the development of a machine-learning model for the recognition of O. viverini eggs in digital images. However, some issues regarding the development of the model must be considered before publication. The authors recognize the need for improvement in the model so that it can be applied clinically and the main limitations of the study in the Discussion section. They also highlight the importance of the sample preparation technique as an important factor for the acquisition of good-quality images for diagnosis and model development.

Experimental design

With the small number of images, the authors highlight the use of data augmentation as the main consequence of the good performance of the model. However, the good performance is the result of overfitting. I suggest that authors evaluate the trained model after 5-fold cross-validation with 8:1:1 ratio (train:validation:test) to fit the small number of available images (For more details see: Rodriguez et al at https://ieeexplore.ieee.org/abstract/document/5342427). Another experimental failure consists in not presenting the results referring to the data augmentation. It is necessary to demonstrate that a model trained with data augmentation performs better than a model without data augmentation. Specifically in instances of image rotation, since CNN models are not invariant in the rotation or scale of objects.

Another indicator that the model is overfitting is the large number of epochs. Figures 8 and 9 show that there is no improvement in model performance after 50 epochs. To avoid overfitting, the model should be trained for an optimal number of epochs using an early stop function.

Validity of the findings

Although the authors present the raw images used to test the model, it is not possible to verify if the images used in the test set are the same as those used to create the training and validation set. There are 148 images available for testing, however, the images that were used before pre-processing for training are not available. Regarding the model, the code is available, but it remains to describe some details of the model so that we can reproduce the analyses.

Additional comments

I have placed additional comments in the attached document.

Annotated reviews are not available for download in order to protect the identity of reviewers who chose to remain anonymous.

Reviewer 2 ·

Basic reporting

The authors suggested original technical application for diagnosis of parasite infection. The problem of Opisthorchis egg detection is important for the opisthorchiasis disease diagnostics. The problem has complex character. The reporting is quite detailed. It fits to the journal standards.
The illustrations are complete. The language and presentation style are clear.
The manuscript is well Illustrated. Even some figures might be redundant.

Experimental design

Sample preparation and computation technique is properly described.
May add some words about YOLO method series.
There are several versions of the algorithm as well as multiple CNN variants.
Need mention demands for hardware used for the model training.

Validity of the findings

I have some remarks on the references. It is good to describe opisthorchiasis as the disease in greater details, underline its importance for other regions, not only South-East Asia. I mean Opisthorchis felineus and similar problems of diagnostics by detection of small eggs in stool. Thus some discussion extension as perspectives of the method extension.
The prediction results (accuracy and classification metrics) are good. Need show it separately for the training and control sets.

Additional comments

Line 31: ‘Human opisthorchiasis.. is caused by … Opisthorchis viverrini’ – please add a phrase that opisthorchiasis is dangerous chronic diseases distributed in many Asian areas in water-basins of large rivers, in Siberia, as well as in Europe.
Line 45: ‘dataset from 1x to 36x ...per class’ – wording ‘1x’ is not clear, maybe write strait ‘in 36 times’?
And ‘per class’ might be misunderstand that second class (eggs) is 36 times larger than the artifacts class.
Compare with phrase in line 51, change there wording too.
Line 53: ‘area under the curve equal to 1.0.’ – may write ‘area under the curve’ as the standard ROC-AUC term. The value ‘1.0’ is perfect, note again that this accuracy is reached on the training set.
Line 66: - add a phrase and common references about opisthorchiasis, its distribution worldwide and the causing fluke species.
Mention Opisthorchis felineus, it is similar species with same problems of detection:
May add references to these publications
https://pubmed.ncbi.nlm.nih.gov/32598389/
https://pubmed.ncbi.nlm.nih.gov/26838224/
https://pubmed.ncbi.nlm.nih.gov/2755409/
Line 78: ‘the infection is associated with several hepatobiliary diseases, including cholangiocarcinoma’ – add here references for these associated diseases.
Line 85: ‘Typically, suspected cases are required to provide stool samples for microscopic examination.’ – please add here references to other diagnostics – such as immune-tests,PCR and expected accuracy.
May cite these work for current protocols:
https://pubmed.ncbi.nlm.nih.gov/1852493/
https://pubmed.ncbi.nlm.nih.gov/9377342/

Please avoid citing in bulk, more than 3 references together –
See line 100-101:
‘drug discovery, and disease diagnostics (Topol, 2019; Cai, Gao & Zhao, 2020; Kumar & Mahajan, 2020; Sun et al., 2020; Vaisman et al., 2020).’ –
Put each reference to proper place in the sentence – what is for drugs, what is for diagnostics. Add details on other diseases. It will extend the readers ‘interest
Lines 109-111: bulk references. Cite separately, each for specific type of study.
Line 113: ‘YOLOv4-Tiny’ – need write in short about YOLO models. Mention progress in the model development. Recent applications are based on YOLOv7
Line 124: ‘use in remote regions of developing nations’ – it is not about developing countries – the tool is effective everywhere. Need describe technical features and hardware demand – what kind of computer, operation system is necessary, if high-throughput computing was used for this task.
The laboratory testing based on egg detection in feces still may miss the disease. There is no 100% effectiveness, as it was noted in literature. Combination of testing methods and sample preparation protocols was recommended previously.
Line 165: ‘These images were carefully prepared’ – here word ‘carefully’ may mislead. Sample preparation should correspond to standard protocols, no special improving or manipulation with the data. Assume it is part of algorithm training, not data processing.
Line 182: ‘we employed four kinds of filters’ – please add a reference, why these four methods, or it is following to previously published protocol.
Line 210: ‘Rectified Linear Unit (ReLU)’ – add reference to this term. Where ReLU was used previously?
Line 229 and the paragraph below: “recall, accuracy, and the F1-score, which are defined as follows:”
Description of the standard metrics is not necessary. May write it shorter, add a reference to the standards.
Line 355: The terms ‘(sensitivity)’ and ‘(1-specificity)’ could be given in parentheses, as the parameters. Or use standard abbreviations like ‘sn’, ‘sp’.
Lines 428-431: ‘ deep learning in the field of parasitology diagnostics…’ – bulk citation. Too many references together – add details, cute separately, name the application fields.
Line 439: ‘The application of more advanced machine learning methods should be investigated for future studies.’ – this phrase is too common. Name other methods, state the problem. Currently it is 100% accuracy. What to improve a nd why?
Assume it is for larger datasets, sample preparation protocols, other types of parasite eggs.
Line 449: ‘telehealth applications..’ – again bulk reference.
Problem of distant consulting, and remote access to laboratory analysis from field testing is interesting. It is worthy extend the discussion.
Line 469: ‘…F1-score of 1.0’ – all scores were 1 (maximal)? Might rephrase here.

Legend for figure 3: ‘It consists of a convolutional layer, a max pooling layer..’
There are several convolutional and pooling layers in the figure. Either mark them in graphics, or change figure legend.
Figure 10. The Receiver Operating Characteristic (ROC) curve.
It is 1.0 line (100% full). Line is not visible. Please note that it is good for the training set
Maybe not show this figure.
Or show additionally ROC curve on the control set. It’d be visible, slightly lower than ideal lint on the top.

Reviewer 3 ·

Basic reporting

The authors present an approach that improves the reproducibility and specificity of egg recognition on microphotographs of fecal samples. Considering that the counting eggs in feces using manual and subjective analysis is still the main worldwide diagnostic tool for liver fluke detection in patients, this improved approach is certainly useful for laboratory and field diagnostics and will improve and facilitate the manual method of testing eggs in fecal samples.

Experimental design

Methods described in details and supporting information is sufficient.

Validity of the findings

Statistical approach is relevant. The data statistically sound.

Additional comments

However, some species of helminths produce eggs of the same shape and size, and it is not possible to distinguish one species from another, in particular, this applies to trematodes of the Opisthorchidae family. Some endemic areas for O. viverrini and C. sinensis overlap, and in these areas it is possible that diagnostics will demonstrate infection not only with O. viverrini, but also with Clonorchis. In my opinion, the manuscript focuses only on the diagnosis of O. viverrini in Thailand. Although the authors briefly mention the limitations of their approach in the text of the Discussion section, in my opinion, it is not sufficient. Expansion of the discussion on the limitations of the method in relation to testing other species would improve the text of the paper.
Moreover, the whole text is too concentrated on one species of parasites and only in the local territory of Thailand. I recommend expanding the background and discussion also with information on other parasite species and similar automated approaches for diagnosing epidemiologically significant human parasites. In this case, the article will be of interest to a wider readership.

---

## Round 0.2 · Minor Revisions

The authors addressed the main concerns of the reviews. However, the revised manuscript still deserves attention. Please, provide point-to-point responses according to the comments made the Reviewer #1 in the new version of your manuscript.

Reviewer 1 ·

Basic reporting

No comment.

Experimental design

The authors incorporated the suggestions into the manuscript. However, I still suspect that the developed image classification model is overfitting. Looking at the code, we can see that the data augmentation process is applied to all 200 images in the training set. On the same set of images, images were separated to test the classifier model.

I recommend applying image augmentation techniques to the training and validation sets while retaining the test set with only the original images (https://arxiv.org/abs/2207.03342). Alternatively, the application of data augmentation techniques can be done on the test data set, after the images have been separated, to verify the robustness of the model (https://doi.org/10.1186/s40537-019-0197- 0). These processes can be incorporated into the model sequence for training and validation (https://keras.io/api/keras_cv/layers/augmentation/).

Validity of the findings

It is important to highlight that the results presented show the performance of an image classifier model, and the observed sensitivity/specificity does not reflect the sensitivity of a diagnostic method. To assess the sensitivity of a computational model as a diagnostic tool, it is necessary to carry out a clinical study with a larger cohort, compare it with the diagnostic method considered the gold standard, and take into account the prevalence of the disease in question in the population studied.

Despite the excellent performance presented by the image classifier model, the authors did not present a comparison with other models such as AlexNet (https://arxiv.org/abs/1404.5997), VGG-16, VGG-19 (http://arxiv.org/abs/1409.1556), ResNet (http://arxiv.org/abs/1512.03385), Xception (http://arxiv.org/abs/1610.02357), or DenseNet (http://arxiv.org/abs/ 1608.06993). The proposed model must be compared with other existing models.

The authors have not yet presented an evaluation of the model on object detection. Despite the good performance in correctly classifying images, it is necessary to properly present performance in detecting objects in images. The algorithm performs a patch search and applies the classifier model. Only the degree of confidence is presented as a metric for detecting objects (eggs or artifacts in patches with a probability greater than 80%) in the images. However, the "heatmaps" presented do not show the probability map of the objects, nor the average precision (AP@50) obtained in detecting each instance (object in the image). Heatmaps can be obtained by applying Grad-CAM (https://link.springer.com/article/10.1007/s11263-019-01228-7). I suggest the authors also present a figure of the heatmaps on the images (overlay) to see which patches and their respective degrees of confidence. This way one can make a visual error analysis of the model.

Additional comments

Figures 4, 5, and 6 can be joined into a single figure to show the model training and validation.

Figures 8, 9, and 10 can be joined into a single figure to show how the model performs on object detection.

Reviewer 2 ·

Basic reporting

This is second review for this manuscript. It highlights new machine learning approach for Opisthorchis viverrini egg classification. The manuscript has been updated according to my suggestions. I have no more comments.

Experimental design

Experimental design is appropriate for this analysis.

Validity of the findings

I have no more comments.

Additional comments

I have no more comments.

---

## Round 0.3 · accepted · Accept

The authors have addressed the primary concerns raised by Reviewer #1. When responding to the list of production tasks, kindly take into consideration the formatting points highlighted by Reviewer #1.

Reviewer 1 ·

Basic reporting

The authors have addressed all the questions. I have no more comments.

Experimental design

I have no more comments.

Validity of the findings

The authors have addressed all the questions. It should therefore be highlighted under which conditions the computational model can be used. Additionally, please add the classification results of the other models (AlexNet, VGG-16, ResNet, etc.) evaluated in the ROC curve.

Additional comments

To improve readability, please improve the resolution and increase the font size of Figures 4 and 5.